# Wheat Farmers' Perception of Constraints and Their Adaptive Capacity to Changing Demands in Egypt

Ahmed Abdalla [1,*], Till Stellmacher [1] and Mathias Becker [2]

1    Right Livelihood College (RLC), Center for Development Research (ZEF), University of Bonn, 53113 Bonn, Germany
2    Institute for Crop Science and Resources Conservation (INRES), University of Bonn, 53113 Bonn, Germany
*    Correspondence: s5adabda@uni-bonn.de

**Abstract:** Most of the approximately 105 million Egyptians depend on wheat in the form of *baladi* bread for their daily diet. Millions of smallholders along the River Nile have produced wheat for millennia; however, in more recent history, the wheat demand and supply ratio has dramatically changed. The first wheat imports in Egyptian history were in 1966. Today, domestic production meets only half of the wheat consumption, and Egypt has become the largest wheat importer in the world. Before the Russia–Ukraine war, 85% of the wheat imports to Egypt came from Russia and Ukraine. The war and the associated disruption of the wheat supply chains has put Egypt on the top list of so-called "developing countries highly threatened by food crises". Against this backdrop, we analyzed decision-making factors and perceptions of wheat-producing smallholders in the Nile River Delta, the wheat basket of Egypt. The study draws on nine months of empirical fieldwork in the Nile River Delta. We employed a mixed approach to data collection, combining interviews and focus group discussions with smallholders, experts, and agriculture extension agents with transect walks and field observations. In total, 246 randomly selected wheat-growing smallholders were interviewed in four divisions in the Nile River Delta. Our findings show that the production of wheat by smallholders is highly influenced by system-immanent factors, such as subsistence need for home consumption and the presence and intensity of animal husbandry, as well as by external factors, such as the domestic prices for wheat determined by the government in each season and the time of the declaration of these prices. These factors affect smallholders' decisions to increase or decrease their wheat cultivation area. However, the study also showed that the factors influencing farmers' decisions to grow wheat or implement innovative practices vary across different areas within the same region. Smallholders struggle with poor access to fundamental production factors and are discontented with the low provision of extension and support services as well as poor market structures. These constraints act as disincentives for smallholders to produce (more) wheat. They need to be addressed and eliminated to increase domestic production and to reduce Egypt's dependency on expensive and unreliable wheat imports.

**Keywords:** food security; wheat self-sufficiency; Egypt; smallholders; agricultural policies





## 1. Introduction

In many parts of the world, smallholder farmers are the backbone of agriculture and for food security. Smallholder farmers produce 50 to 80% of the world's food and up to 90% in the Global South [1,2]. Smallholders are defined as farmers producing on relatively small land holdings, mainly with family labor, and using parts of the production for their own household consumption [3]. Agricultural productivity, sustainability, and agricultural transformation largely depend on the decisions of smallholders to grow certain crops at a given time and quantity. However, the process of decision making at the smallholder level is complex and influenced by many factors. Tanvi [4] grouped those in physical factors, economic factors, farmer preferences, crop profiles, and the availability of resources.

Talawar [5] conducted a study on smallholders' decision making based on physical factors such as soil quality and the availability of water. Other studies show crop attributes such as a resistance to pests and diseases, growth cycles, and fertilizer requirements as important factors influencing smallholders' decision making, while other authors stress the role of the availability of extension services and inputs such as farm machinery, fertilizers, and pesticides [5]. Morgen [6] considered economic factors as fundamental for smallholders' decision making, stating: "Essentially, farms are businesses with economic objectives". On the other hand, Briggs [7,8] showed that economic factors, such as projected market prices and crop production costs, play smaller roles in the decision-making processes of smallholders. Greig [9] attributed a "limited commercial nature of smallholder agriculture" and found that smallholders' decisions to produce a certain crop at a given time and quantity are influenced by factors such as workload, experiences and traditions rather than by purely economic considerations.

### 1.1. Smallholders in Egypt

In Egypt, 90% of all farmers are smallholders with less than 2 ha of cropland; 50% of all farmers in Egypt have even less than 0.4 ha [1,10]. These farms employ a mix of traditional and modern agricultural practices, adopting technologies such as improved seeds and mechanization to enhance wheat cultivation. Average wheat yields on these farms range from 3 to 6 tons per hectare, but with better access to agronomic practices, higher yields of up to 8.5 tons per hectare are achievable. Inputs like wheat seeds, fertilizers, and pesticides are procured from local supply stores in the private market or cooperatives, with government subsidies. However, challenges exist in ensuring timely and sufficient supply [11]. Smallholder farming is the backbone of agriculture and for food security in Egypt. It contributes significantly to livelihoods and employment in rural areas, to land and water use, and to social belonging and cultural heritage of large parts of the population. However, policies of different Egyptian governments over the past decades led to a reduction in agricultural subsidies and extension support for smallholders, incentivizing large-scale industrial production of export crops, rather than staple food production, and generally to a disaggregation and reconstitution of the smallholder's class in Egypt.

### 1.2. Wheat Importance in Egypt

The population of Egypt is growing at 2.2% per year [11]. It is expected to reach 157 million people by the year 2050 [11]. Projections indicate that the demand for wheat will double by 2050 [12]. Demographic growth combined with dwindling support for smallholders will further increase the reliance on imported wheat [13]. On the other hand, policies have supported the expansion of agriculture to land in the desert (the "new lands"), but with a focus on export-oriented commodities such as citrus. Over the years, Egypt not only became an important exporter for certain agricultural products; it also became the largest wheat-importing nation in the world [14]. Wheat is a strategic commodity and historically the most important crop in Egypt. Egyptians derive one-third of their daily caloric intake and 45% of their protein intake from wheat-based food [15], mainly in the form of bread called *aish baladi* (*aish* means "life" in Arabic). The political stability in Egypt is closely linked to the price and availability of *baladi* bread. For example, the rise of the prices for *baladi* bread was the main reason for sparking the Egyptian "bread riots" in 1977. Thirty-four years later, the prices of *baladi* bread also played a critical role in the Egyptian revolution in 2011, where the call of the revolution was "bread, freedom, and social justice".

After decades of political negligence of wheat-producing smallholders on the one hand, and population increase of about 2 million per year on the other hand (Abdalla, et.al., 2022), only less than half of the national wheat consumption can be met by domestic production; the rest has to be covered by wheat imports [16]. In 2018, from the national wheat consumption of 18.5 million metric tons, about 12 million tons had to be imported [17]. In 2020, Egypt had a total wheat consumption of 20 million metric tons of which 13.5 million were imported [18,19]. In recent years, Russia and Ukraine were by far the most important

wheat suppliers to Egypt. About 60 to 66% of the wheat imports to Egypt (depending on the years) came from the Russian Federation and another 20 to 25% were imported from Ukraine. This was followed by smaller amounts from Romania, France, and the United States [19]. In 2021, briefly before the war, Russia and Ukraine together contributed 85% of the total wheat imports to Egypt [18,19].

About 61% of the Egyptian population (63.5 million people) from the total population of about 105 million, especially the poorer ones rely on *baladi* bread for their day-to-day diet. *Baladi* is sold under a state-subsidized food card system [19]. The government fixed the price of *baladi* at EGP 0.05 per loaf (equivalent to USD 0.01) in January 2022, which is less than one-tenth of the actual production costs [17]. This bread subsidy program has become a massive economic burden on the Egyptian state budget. For instance, in the fiscal year 2019/20 alone, the government allocated more than EGP 89 billion (USD 5.69 billion) for bread subsidies. In the fiscal year 2020/21, bread subsidies amounted to EGP 84.5 billion (USD 5.4 billion) [20]. Most of these subsidies targeted the wheat supply chain, from buying wheat from producers, to flour milling, via bread production in bakeries, to selling it to the end consumers [21]. They, however, largely neglected support for wheat-producing smallholders.

### 1.3. The Impact of the Russia–Ukraine War on Egypt

In 2022, the value of the Egyptian currency dropped by about 60% against the USD (EGP 15.7 to USD 1 in January 2022 compared to EGP 24.7 to USD 1 in December 2022) [22]. The currency devaluation was partly a result of the massive trade deficit of the Egyptian economy. This faced markets in Egypt with a disruption in the supply chains of several essential commodities, in addition to a significant price hike for food coupled with a lack of foreign currencies [23].

The recent UNCTAD report [18] highlighted that many countries in the Global South that rely on food imports, and Egypt specifically, are facing a double burden of massively increasing prices for food imports and a depreciation of their currencies against the USD. This increases the risk of hunger for millions of people, particularly the poor, in these countries. In October 2022, the wheat prices in the international market increased by 89% [18]. Additionally, the average exchange rate between the USD and respective national currencies in food import-depended countries of the Global South increased by 10–46%. Combined the real food import prices increased between 106 and 176% [18].

The UN-CTAD report showed that exchange rate effects are significant drivers of rising food import bills, contributing to inflation, loss of purchasing power and food insecurity in the impacted nations. After Mauritius, Pakistan, and Ethiopia, Egypt is the fourth-most affected country in the world by these exchange rate effects. For instance, the wheat import bill in Egypt increased by 112% between 2020 and 2022 due to the combined effects of increased wheat prices on international markets and currency devaluation. As a result, to import the same amount of wheat as in 2020 (13.5 million metric tons), the additional costs are estimated at USD 3 billion in 2022, which is equivalent to 20% of the total expenditures for food import in Egypt [18]. This problematic wheat import situation jeopardizes national food security, particularly for the poorer parts of the population. Despite a World Bank loan of USD 500 million in June 2022 [19] to improve the access to bread for poorer households through the Emergency Food Security and Resilience Support Project, food insecurity is dramatically increasing and threatening the social-political stability in Egypt.

Already in 2014, the Government of Egypt issued the vision of substituting costly and increasingly insecure wheat imports by increasing domestic wheat production in its "Egypt Sustainable Development Strategy Towards 2030". The strategy was aimed at increasing domestic wheat production and attaining a wheat self-sufficiency level of 74% by 2017. However, in reality, wheat self-sufficiency in 2017 was only 43% [14].

*1.4. Study Objectives*

Against this background, this study aims to determine the main challenges facing domestic wheat production in Egypt by:

(1) Determining the key factors influencing smallholders' decisions to grow wheat and adopt efficient cultivation practices;

(2) Showing the perspectives of wheat-growing smallholders on their (dis-)incentives to increase or decrease their wheat production; and

(3) Evaluating changes in key determinants, influencing smallholders' decisions over two decades (2000–2020).

(4) Testing our hypothesis that the factors impacting their decision-making process differ across the four study regions due to varying levels of access to water, labor, market opportunities, and soil types.

The research focused on four municipal divisions in the Nile River Delta of Lower Egypt, which is traditionally the main wheat-growing area and the "breadbasket" of the country.

## 2. Methods

*2.1. Description of the Study Area*

About 57% of the wheat-producing land in Egypt is located in the Nile Delta [24]. Beheira Governorate covers 9826 km$^2$ west of the Rosetta branch of the River Nile. The area is densely populated by about 7 million people, of which more than 70% are working in agriculture. The area is well connected by four highways to the central markets of Cairo and Alexandria. It consists of 13 centers, 14 cities, 84 rural municipalities, and 407 villages [25]. While representing only 15% of the total agricultural land in Egypt [26], Beheira Governorate produces 60–65% of the total wheat of the country. Spring wheat is grown in rotation with irrigated summer crops and vegetables on relatively small land holdings in the peri-urban fringes of the "old lands", and with partial pivot and spray irrigation in fallow rotation systems on large land holdings in rural areas of the "new lands" [27]. Four municipal divisions were selected as study areas, based on their diversity in water availability, market access and soil type. The research transect extends between (1) Al Mahmoudiya Division, consisting of 14 villages located on the Rosetta branch of the Nile (rural–moist scenario), (2) Kafr El Dawar Division, consisting of 31 villages located along major highways with very good market access (urban–moist scenario), (3) Abu El Matamir Division, consisting of 24 villages located on the desert margin (rural–dry scenario), and (4) El Nubariyah Division, representing the cultivated desert (urban–dry scenario) (Figure 1).

Within each of the study divisions, different wheat production and marketing methods have evolved over time in response to different external pressures (policies, water availability and pricing, input, and product process) and system-immanent drivers (the availability of production factors and farmers' perceptions) that were systematically characterized and categorized [14].

*2.2. Data Collection*

Using a mixed-method approach, we collected both qualitative and quantitative data from four municipal divisions in Beheira Governorate of Lower Egypt. Qualitative data were collected through 28 open interviews with wheat-growing smallholders, 12 with agricultural extension agents, and 8 with agricultural cooperative representatives. In addition, we combined 12 focus group discussions (FGDs) with participatory observations and transect walks. To collect quantitative data, we selected 246 wheat-growing smallholders in four municipal divisions of Beheira Governorate, namely Al Mahmoudiyah, Kafr El Dawar, Abu El Matamir, and El Nubariyah, using random sampling methods. Interviews were conducted during the 2020/2021 agricultural season using a pre-tested questionnaire. The distribution of the sample (number of smallholders) was proportional to the respective number of farms in each division (corresponding to 0.6% of its total number). The sample

size was sufficient to achieve the desired level of statistical precision (95% confidence interval with a margin of error of ±5%). Moreover, the sample size allowed for subgroup analyses and comparisons across different variables of interest, such as farm size and farming experience, providing a robust basis for drawing meaningful conclusions. The perceptions of wheat-growing smallholders were assessed using a five-point Likert scale. The Likert scale allowed the interviewees to indicate (1) the extent to which they agreed or disagreed to certain statements related to wheat production, (2) how they assess certain external pressures and system-immanent drivers (very poor = 1, poor = 2, acceptable = 3, good = 4, excellent = 5), and (3) how they perceive changes of the same between 2000 and 2020 (much worse = 1, worse = 2, same = 3, better = 4, much better = 5). This provided a structured and standardized approach to data collection, allowing comparisons across different groups of respondents.

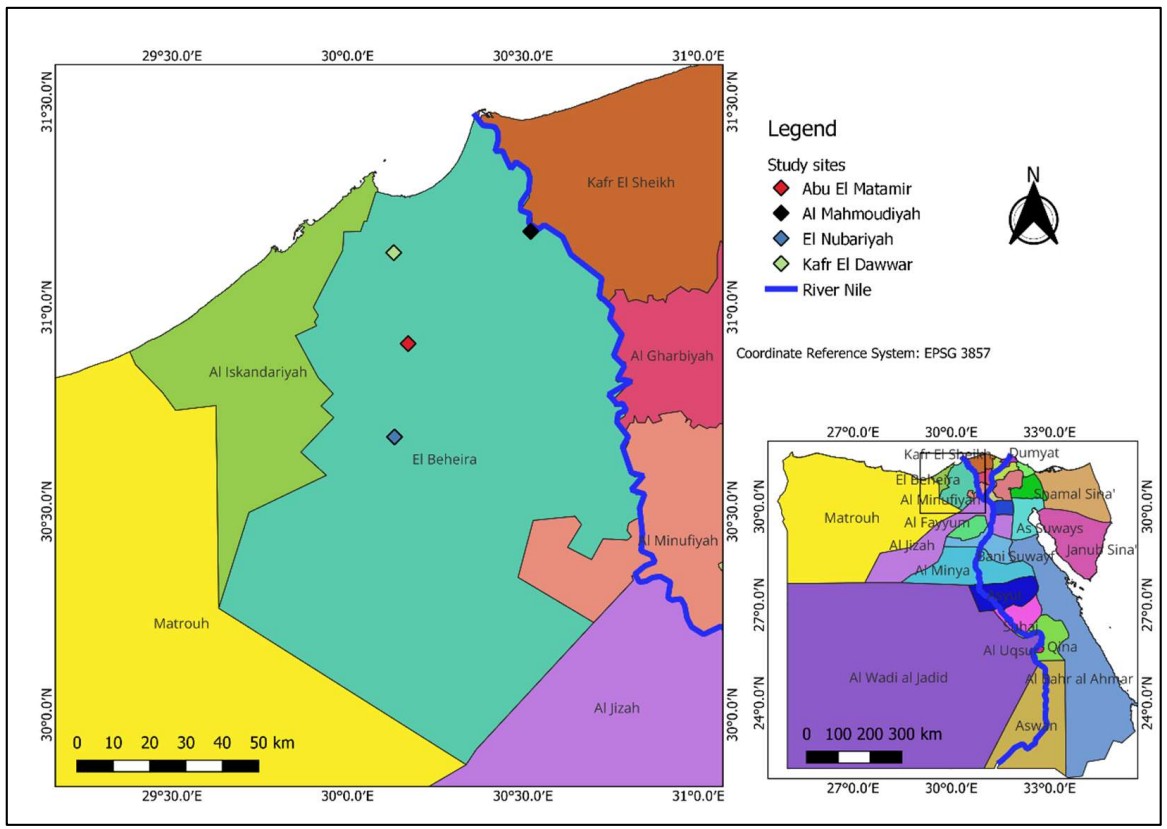

**Figure 1.** Study locations (municipal divisions) within El Beheira Governate in northern Egypt. Al Mahmoudiya, Kafr El Dawar, and Abu El Matamir Divisions represent the "old lands", while El Nubariyah Division represents the "new lands".

*2.3. Data Analysis*

2.3.1. Qualitative Data Analysis

The qualitative information empirically collected from various sources, including 28 open interviews with wheat-growing smallholders, 12 interviews with agricultural extension agents, 8 interviews with agricultural cooperative representatives, and 12 focus group discussions (FGDs) were triangulated and combined with participatory observations and transect walks to capture a comprehensive understanding of the research topic.

The process of data analysis followed a systematic and iterative approach. Initially, the interviews were transcribed verbatim, and detailed field notes were documented from the participatory observations and transect walks. This ensured a comprehensive record of the collected data. The data were then coded, and we carefully examined the transcripts, field notes, and other data sources to identify meaningful units of analysis, such as phrases, sentences, or paragraphs, and assigned descriptive codes to capture emerging themes and

patterns. The coded data were subjected to a rigorous analysis to identify and develop themes that encapsulated the underlying patterns and trends present in the dataset.

Triangulation was employed to enhance the deepness and trustworthiness of the findings. We examined the convergence and divergence of findings across different data sources, such as interviews, FGDs, and observations. This comprehensive approach allowed for a more nuanced understanding of the research topic and provided opportunities to explore complex variations in perspectives and experiences.

Throughout the analysis, relevant quotations and excerpts from the data were meticulously selected to add background information and/or to provide supporting evidence. These excerpts were chosen to enrich the analysis with concrete examples and participant voices. By including these quotations, the study ensures more transparency and credibility of the findings.

The final stage of the qualitative data analysis involved synthesizing and reporting the key findings. The identified themes, subthemes, and supporting evidence were synthesized into a coherent narrative, highlighting the most salient insights derived from the qualitative data. The findings were directly related to the research objectives to deepen the understanding of the research subject.

### 2.3.2. Quantitative Data Analysis

The analysis of the quantitative data involved employing descriptive statistics, such as frequency counts, percentages, and means, using STATA software. These statistics were utilized to describe wheat production characteristics and to assess the perceptions of wheat-growing smallholders.

To gain a deeper understanding of the dataset's structure and to identify important factors, a Principal Component Analysis (PCA) was performed. The PCA allowed for the transformation of a high-dimensional dataset into a lower-dimensional representation while preserving essential information. This phase aimed to identify the key factors that influence wheat-growing smallholders decision making by applying PCA to Likert scale data. The dataset consisted of 246 samples with 10 variables, which captured various attributes of smallholders and their farms.

In addition to the PCA, a linear regression analysis was conducted to explore whether significant differences existed between the PCA factors and the four study areas. This analysis aimed to assess the relationship between the identified PCA factors and the geographic study areas.

## 3. Results

### 3.1. Key Production Characteristics of Wheat-Growing Smallholder Farms

The irrigation in the "old lands" in El Mahmoudiya, Kafr El Dawar, and Abo El Matamir divisions mainly works through pump surface irrigation from canals. In contrast, the wheat irrigation in the "new lands" in El Nubariyah division is mainly a spray irrigation system. The summer-winter rotations on the clay soils in El Mahmoudiyah and Kafr El Dawar divisions are mainly maize–wheat and rice–wheat. However, in Abo El Matamir division with clay loamy soils, potatoes-wheat, maize-wheat, sunflower-wheat rotations are more common. The summer–winter rotations in the sandy soil of Nubaria division are peanuts–wheat, maize–wheat, potatoes–wheat, and sesame–wheat. The survey conducted with 246 wheat-growing smallholders showed the following socio-economic characteristics as shown in Table 1.

The results reveal that wheat-growing smallholders have a mean age of 56 years, with a minimum of 32 years and a maximum of 79 years, and a mean experience in wheat cultivation of 30 years. The smallholders hold land with a mean of 1.1 ha, with a minimum of 0.2 ha and a maximum of 2.3 ha. The mean of the wheat yield is 6.4 t/ha, with a minimum of 4.2 t/ha and a maximum of 8.5 t/ha.

**Table 1.** Key characteristics of wheat-growing smallholder farms (source: field survey 2020/21 growing season).

|  | Age of Farmer (Years) | Farming Experience (Years) | Farm Size (ha) | Wheat Yield (t/ha) |
|---|---|---|---|---|
| Mean | 56 | 30 | 1.05 | 6.4 |
| Range | 47 | 40 | 2.1 | 4.2 |
| Minimum | 32 | 10 | 0.2 | 4.2 |
| Maximum | 79 | 50 | 2.3 | 8.5 |
| Count (n) | 246 | 246 | 246 | 246 |

*3.2. Factors Influencing the Decision of Smallholders to Grow Wheat*

Both results from the open interviews and questionnaire-based interviews determined ten key factors influencing the decision of smallholders to grow wheat. The influence of these factors was evaluated and ranked based on smallholders' perspectives, as shown in Figure 2.

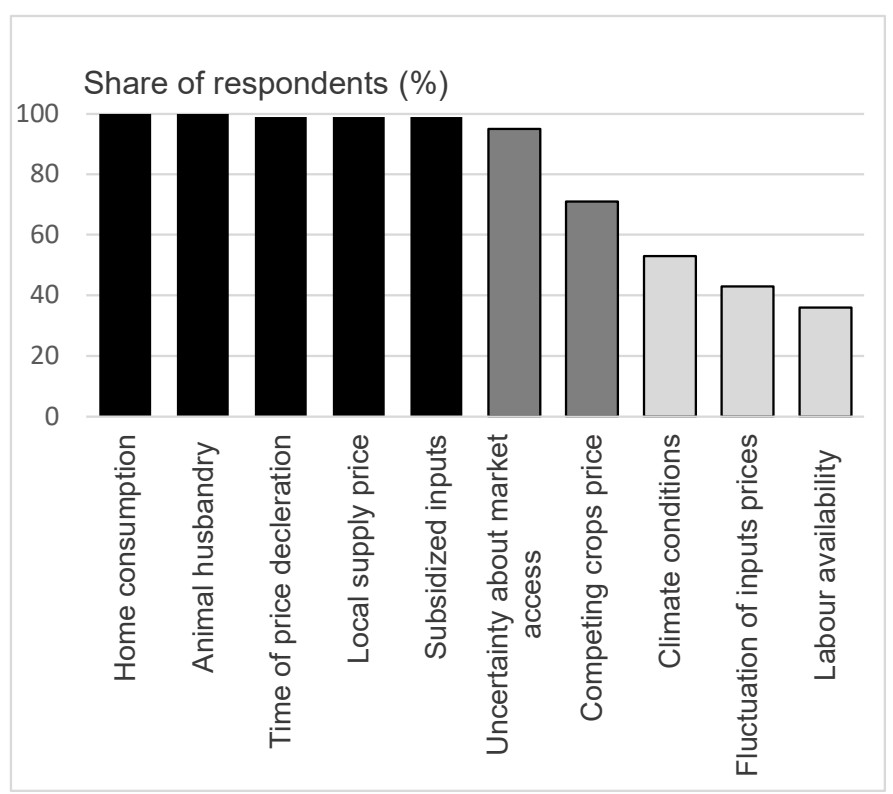

**Figure 2.** Key factors influencing the decision of smallholders to grow wheat (source: field survey 2020/21 growing season).

In all cases, the household consumption of wheat (subsistence needs) and animal husbandry (crop residue uses and forage production) determined smallholders' decisions to grow wheat as system-immanent drivers. In 99% of all cases, farm-external determinants, namely the time of the price declaration by the Ministry of Supply and Internal Trade, the local supply price and the availability of subsidized inputs, influenced the decisions of smallholders. In 95% of the cases, uncertainty about market access influenced the decisions. Other external factors included the prices of competing crops (71%), climatic conditions (53%), fluctuation of input prices (43%), and labor availability (36%).

*Household consumption* refers to the share of the agricultural production used for own consumption to secure households' needs. Smallholders with large subsistence needs tend to produce wheat for their own consumption. However, profitability can significantly

influence decisions of increasing or decreasing the share of land on which they grow wheat, as highlighted in the below statement of a smallholder:

> "Wheat is an essential crop for us as most of our food is wheat-based. I own 1.5 hectares, and 80% of my land share is for clover production and 20% for wheat, but if the wheat production provides higher revenue, I would increase the share of wheat in substitution for the clover share area." *Smallholder, FGD No. 1*

*Animal husbandry* refers to the livestock smallholders' own. This influences the need for wheat straw as animal feed or as bedding material. Thus, smallholders owing livestock perceive wheat straw as an important by-product of wheat production that can be stored for long periods and that can be sold if needed. A statement of a smallholder underlines this:

> "We use the wheat straw to feed our animals and in the case of need, we can easily sell it to our neighbors." *Smallholder, FGD No. 1*

*Price declaration* refers to the time when the Ministry of Supply and Internal Trade declares the local wheat supply price, meaning the price that smallholders obtain for their wheat when selling it to the national public storage silos. In this study, many smallholders mentioned frequent delays by the Ministry declaring these prices. This is a crucial disincentive for them to invest in growing more wheat, or to expand the share of their wheat-growing area. The ministry fixes the local wheat supply prices annually, mostly by the end of the growing season, differentiated in three price levels according to wheat quality and purity. However, most smallholders prefer to sell their wheat to private intermediate traders rather than to public storage silos. They justified this by avoiding transportation costs, waiting lines at the silo gates, and conflicts during the inspections at the silo gates. This is also illustrated in the following statements by two smallholders:

> "Despite the lower price the traders pay to us, I prefer to sell to traders rather than to the public silos. We spend almost one day to sell our wheat to the public silos and in the waiting lines. In addition to the transportation fees." *Smallholder, Interview No. 1*

> "Most often the amount of wheat surplus we intended to sell after saving our home consumption does not deserve the effort of transportation and the time wasted in lines." *Smallholder, Interview No. 4*

Other smallholders criticized the wheat purchasing process at the public silos:

> "The purchasing and inspection process at the public silos is not fair, and in most cases, you need to bribe the inspector to avoid the devaluation of the wheat quality." *Smallholder, FGD No. 2*

> "The inspectors sometimes devalue the wheat to grade three which is approximately EGP 100 lower than grade one for each *ardab*." *Smallholder, FGD No. 6*

> "Traders always deliver large amounts of wheat, and they have their connections to avoid all challenges at the public silos." *Smallholder, FGD No. 9*

*Local wheat supply price* refers to the yearly price set by the ministry of supply for purchasing wheat from the domestic market.

*Subsidized inputs* refer to the fertilizers sold to smallholders by agricultural cooperatives at lower prices than in the private market. Such subsidies are largely limited to mineral nitrogen fertilizers.

*Uncertainty about market access* denotes to what extent smallholders can be certain about selling their wheat at a given time and price. In the 2016 growing season, the Ministry of Supply and Internal Trade ended the purchase of wheat at the public silos earlier than initially declared, saying that the targeted amount of wheat was achieved [28]. This causes serious problems for many smallholders as shown in the following statements:

> "In the 2016 season I lost all my income from wheat because the government stopped the local wheat supply suddenly ahead of the officially scheduled declared date." *Smallholder, Interview No. 2*

> "In the 2016 season I increased the size of wheat after the Ministry declared in November 2016 the price of local wheat supply is 1300 EGP per *ardab*, and suddenly by the end of the season the Ministry declared a different purchasing price of EGP 420 per *ardab* which was a shock to me and all wheat growers." *Smallholder, Interview No. 9*

> "The government prefers to rely on importing wheat from global markets rather than the local market, and always supports wheat importers. I do not believe that the government wants to attain wheat self-sufficiency. The government fighting against smallholders and siding with importers." *Smallholder, Interview No. 12*

*Competing crop prices* refer to the prices of other winter crops competing with wheat cultivation on the same farm area. The most common winter crops competing with wheat in the study areas are broad beans and Alexandrian clover (*bersim*).

*Climate conditions* refer to the weather events affecting wheat quantity and quality, such as floods and high temperature fluctuations during the growing season. Smallholders highlighted the significant negative impact on wheat yield and quality associated with wheat lodging due to excessive rain combined with intense winds. In addition, sudden heat waves during the growing season enhance infections with wheat rust and can lead to massive yield losses. This is exemplified in the following statements:

> "In the last season I lost almost third of my grain yield due to the intensive rain in combination with strong wind." *Smallholder, Interview No. 12*

> "The machinery harvest process became impossible after wheat lodging and we must harvest it manually which requires longer time, manpower, and higher cost." *Smallholder, Interview No. 8*

*Fluctuation of input prices* refers to the variability in prices for agricultural inputs on private markets such as fertilizers, seeds, machines, and fuel. Despite high price fluctuations in recent years, these hardly affected the share of the land devoted to wheat cultivation, but farmers rather reduced the use of such inputs.

*Labor availability* refers to the provision and costs of agricultural labor for the smallholder farm. We found, however, that labor availability is not one of the main influential factors in smallholders' decision to grow wheat. Most smallholders in the study area rely on family labor during the season and reciprocal help from neighbors for more labor-intensive farm operations.

Our results show that the wheat production of smallholders in Egypt depends on a variety of underlying system-immanent and external factors. While smallholders are the backbone of wheat production in the country, many factors limit their ability and willingness to produce (more) wheat. It is essential to understand smallholders' perspectives and to tackle these factors in a "bottom-up approach" to increase food security, to reduce the country's dependency on wheat imports and unreliable supply chains, and to contribute effectively to the "Egypt Sustainable Development Strategy Towards 2030".

Among the system-immanent factors that determine the share of the land area devoted to wheat production are subsistence food needs and requirements for livestock feeding. This all shows that smallholder wheat farming is not an entirely market-based business activity driven by sheer profit maximization but rather an integrated agricultural practice and conduct of life, deeply rooted in food habits and cultural heritage. This is also highlighted in a statement of a smallholder.

> "Wheat cultivation is traditional. We have been growing wheat for ages and we will not stop growing even though its limited profitability as it is essential to our diets and animal feed." *Smallholder, FDG No. 5*

However, the life and culture in rural Egypt are changing. There are changes in traditions that act as disincentives and may ultimately lead to abandoning wheat production altogether, as expressed in the following statement by an agricultural extension agent.

> "About 10 years ago, we were all backing our bread at home, but since the government started to build bakeries for subsidized bread this culture started to change gradually, and women started to not bake as much as before and become more dependent on the subsidized bread. Therefore, the wheat-cultivated areas started decreasing, as the farmers here did not have another reason to produce wheat. Unless they have animals. Alternatively, farmers started to grow more clover." *Agricultural extension agent, Interview No. 2*

Furthermore, among the 10 variables identified above, PCA analysis found that three principal components explained 81% of the total variance in the data. This implies that a significant amount of information can be captured by considering only three variables instead of the original 10. Two variables, namely animal husbandry and home consumption were excluded from the model as they had a constant value of zero and were deemed as the main drivers for wheat cultivation.

Further exploration of the principal components allowed for their interpretation. The first principal component (PC1) showed strong positive associations with variables related to time of price declaration, local wheat supply price, subsidized inputs, and uncertainty about market access, suggesting that it represents a measure of policy status. The second principal component (PC2) exhibited high associations with variables linked to production elements such as fluctuating input prices, labor access, and climate conditions, indicating a production elements component. Lastly, the third principal component (PC3) displayed significant associations with variables related to the prices of competing crops, representing a market component (Component Matrix[a]).

Based on these findings, it can be concluded that policy, production elements, and market factors are the primary contributors to the variability in the dataset. This understanding will guide our further analyses and help interpreting results obtained from the regression model.

Component Matrix[a:] Extraction method; Principal component analysis (Source: field survey 2020/21 growing season).

| Location | PC1 | PC2 | PC3 |
|---|---|---|---|
| Time of price declaration | 0.967 | | |
| Local wheat supply price | 0.967 | | |
| Subsidized inputs | 0.967 | | |
| Uncertainty about market access | 0.700 | | |
| Fluctuation of input prices | | 0.818 | |
| Labors availability | | 0.798 | |
| Climate condition | | 0.663 | |
| Competing crop price | | | 0.704 |

We conducted a linear regression analysis to investigate whether there were significant differences between the three PCA components and the four study divisions.

The results indicated significant differences between the three PCA factors and the four study divisions. The first PCA factor (PC1), "policy", showed significant differences between El Nubariyah Division (urban–dry "D") and the other three divisions (Table 2). However, no significant differences were found among the other three divisions (Figure 3). In El Nubariyah ("D"), the policy factor is considered the most significant factor impacting smallholders' decisions to grow wheat. However, in Abo El Matamir (rural–dry "C"), the policy factor is considered the least significant factor influencing their decisions (Figure 3).

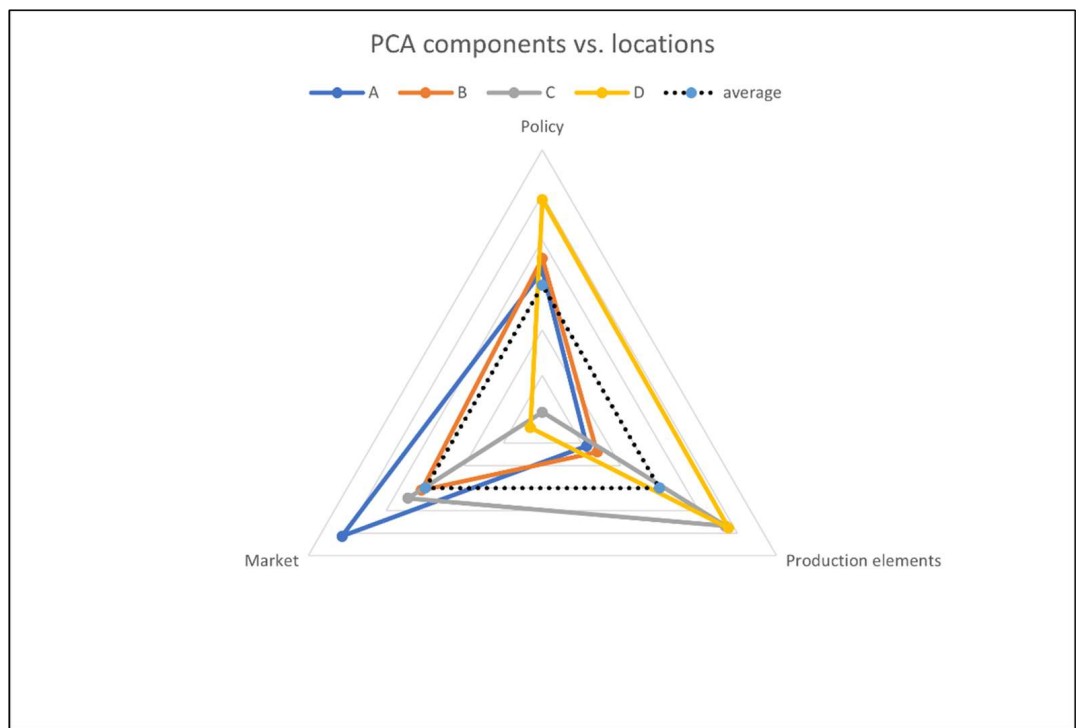

**Figure 3.** PCA factors' variations in the four study divisions (radar chart) (source: field survey 2020/21 growing season).

**Table 2.** Pairwise comparisons of PC1 policy factor vs. study divisions (source: field survey 2020/21 growing season).

| Location | Contrast | Standard Error | t | *p* > t | [95% Conf. Interval] | |
|---|---|---|---|---|---|---|
| B vs. A | 0.0409409 | 0.0346749 | 1.18 | 0.239 | −0.02736 | 0.109244 |
| C vs. A | −0.4734639 | 3443276 | −1.38 | 0.170 | −1.1517 | 0.204797 |
| D vs. A | 0.2378087 | 0.039233 | 6.06 | <0.001 | 0.160526 | 0.315090 |
| C vs. B | −0.5144048 | 0.3447407 | −1.49 | 0.137 | −1.1934 | 0.164670 |
| D vs. B | 0.1968678 | 0.042707 | 4.61 | <0.001 | 0.112743 | 0.280992 |
| D vs. C | 0.7112726 | 0.3452289 | 2.06 | 0.040 | 0.031235 | 1.39131 |

The second component (PC2), "production elements", showed significant differences among the four divisions. However, there were no significant differences between Abo El Matamir ("C"), and El Nubariyah ("D"). Additionally, there were no significant differences between El Mahmoudiya (rural–moist "A") and Kafr El Dawar (urban–moist "B") (Table 3). In the Abo El Matamir division ("C"), the production elements factor (PC2) was considered the most significant driver. However, in El Mahmoudiya, PC2 was found to be the least significant driver (Figure 3).

**Table 3.** Pairwise comparison of PC3 production element factor vs. study divisions(Source: field survey 2020/21 growing season).

| Location | Contrast | Standard Error | t | *p* > t | [95% Conf. Interval] | |
|---|---|---|---|---|---|---|
| B vs. A | 0.0839265 | 0.1312358 | 0.64 | 0.523 | −0.17458 | 0.342436 |
| C vs. A | 1.120849 | 0.1596034 | 7.02 | <0.001 | 0.80645 | 1.43523 |
| D vs. A | 1.146599 | 0.3110838 | 3.69 | <0.001 | 0.53382 | 1.75937 |
| C vs. B | 1.036922 | 0.1685767 | 6.15 | <0.001 | 0.70485 | 1.36898 |
| D vs. B | 1.062672 | 0.3157815 | 3.37 | <0.001 | 0.44064 | 1.68470 |
| D vs. C | 0.0257498 | 0.328585 | 0.08 | 0.938 | −0.62150 | 0.673002 |

Lastly, the market factor (PC3) showed significant differences among all four divisions except between Abo El Matamir ("C") and Kafr El Dawar ("B") (Table 4). In El Mahmoudiya ("A"), the market factor was found to be the most significant factor, while it was the least significant factor in El Nubariyah ("D") (Figure 3).

**Table 4.** Pairwise comparisons of PC3 market factor vs. study divisions (source: field survey 2020/21 growing season).

| Location | Contrast | Standard Error | t | $p > t$ | [95% Conf. Interval] | |
|---|---|---|---|---|---|---|
| B vs. A | −0.577176 | 0.1539015 | −3.75 | <0.001 | −0.8803335 | −0.2740186 |
| C vs. A | −0.4782684 | 0.1451015 | −3.30 | <0.001 | −0.7640915 | −0.1924454 |
| D vs. A | −1.36689 | 0.2222701 | −6.15 | < 0.001 | −1.804721 | −0.9290591 |
| C vs. B | −0.0989076 | 0.1830877 | 0.54 | 0.590 | −0.2617412 | 0.4595564 |
| D vs. B | −0.7897142 | 0.2487382 | −3.17 | <0.001 | −1.279682 | −0.299746 |
| D vs. C | −0.8886218 | 0.2433915 | −3.65 | <0.001 | −1.368058 | −0.4091855 |

In conclusion, the factors that determine smallholders' decision making varied across the four different study divisions, depending on their specific characteristics. For instance, in El Nubariyah, which represents the reclaimed desert "new lands", the policy factor had the most significant decision-making influence (Figure 4). This factor included the timing of the deceleration of the annual wheat price, local wheat supply prices, access to subsidized inputs, and uncertainties regarding market access. This was followed by the production elements factor which played a crucial role in El Nubariyah and can be attributed to its specific characteristics. Notably, the predominantly sandy soil requires additional inputs. Moreover, limited access to water, labor availability, and the absence of substantial animal husbandry practices (a key driver for wheat cultivation) contribute to the shift of many stallholder farmers from cultivating traditional crops like wheat to more profitable alternatives such as vegetables and fruits.

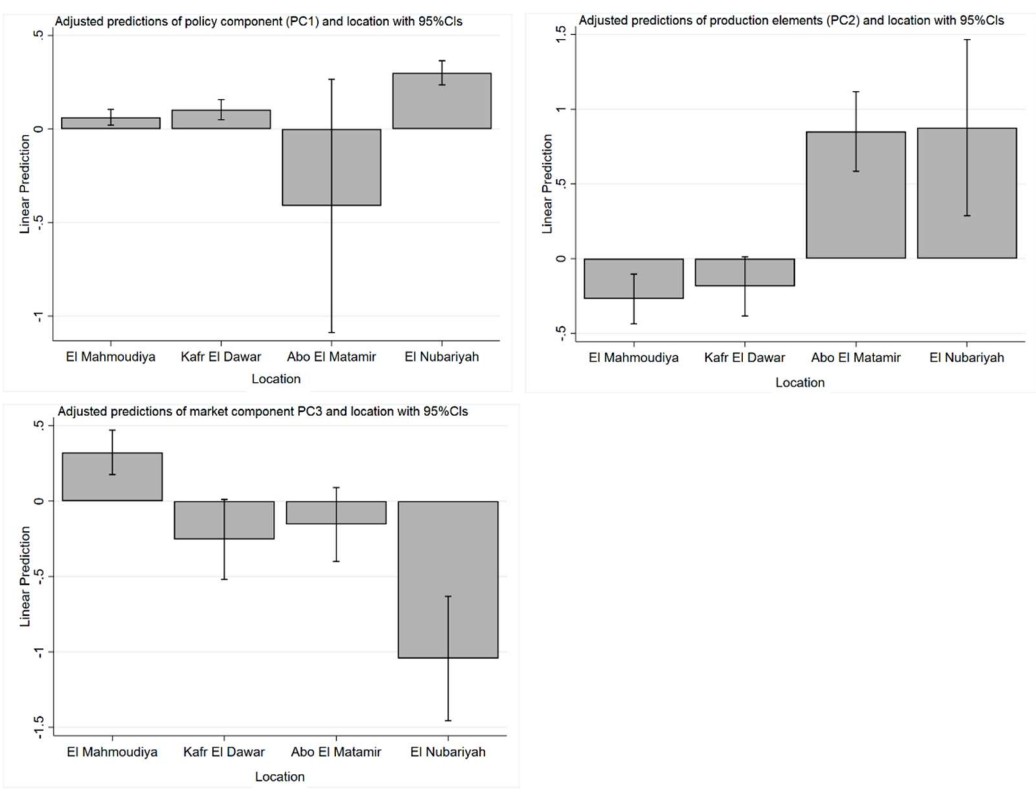

**Figure 4.** PCA factors' variations in the four study divisions (source: field survey 2020/21 growing season).

Conversely, in Al Mahmoudiya, an "old lands" study division characterized as rural–moist, market factors representing prices for competing crops had the most significant influence on smallholders' decision making to grow wheat. This was followed by policy factors in the area. This can be attributed to favorable conditions for crop growth, including the availability of water and nutrient-rich soil (old land), as well as convenient labor accessibility. As a result, wheat cultivation represents only a small proportion (20%) of the land share, with the remaining 80% of the land dedicated to other competitive crops.

### 3.3. Smallholders' Perception of Factors Affecting Wheat Production

The second part of the primary quantitative data collection was aimed at understanding smallholders' perceptions of factors affecting wheat production. Twelve variables were identified from the open interviews and the FDGs, namely: access to subsidized inputs, access to credit, access to agriculture extension services, access to the market, the availability of farm equipment, the availability of agrochemicals, the availability of labor, the availability of water, environmental conditions, input prices, wheat supply prices, and general profitability. Smallholders were asked to rate these variables on a five-point Likert scale from very poor = 1, poor = 2, acceptable = 3, good = 4, to excellent = 5.

As shown in Figure 5, a majority of the smallholders perceived access to the main production factors as inadequate. Nearly three quarters of all smallholders indicated that input prices are very high (74%), while 58% indicated good access to fertilizers on the private market. Furthermore, 53% indicated that the selling price of wheat to be very low, and 40% indicated low. In addition, 52% of the interviewed smallholders indicated that the subsidized input access was very poor, and 43% indicated poor. Overall, the profitability of wheat production was perceived to be very poor or poor (63% and 37% of the interviewed smallholders, respectively).

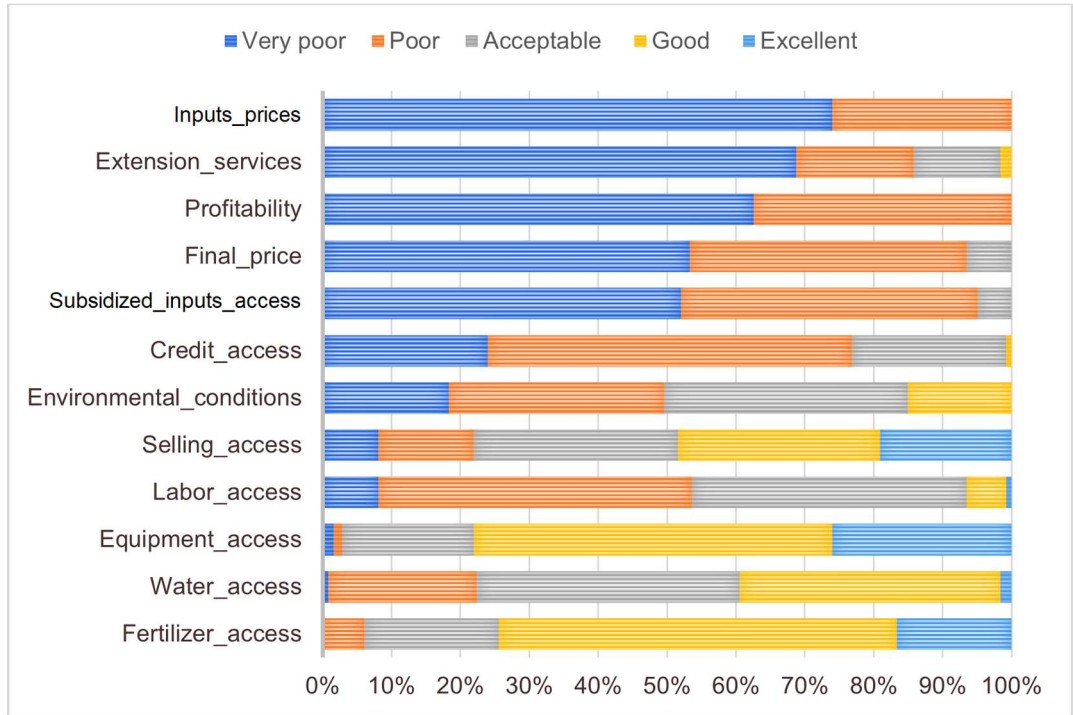

**Figure 5.** Smallholders' perception of factors affecting their wheat production (source: field survey 2020/21 growing season).

Smallholders highlighted that the cost of wheat production inputs significantly increased in the last five years, specifically after the economic reform program in 2016/2017. In particular, the prices of fertilizers increased considerably, and most smallholders cannot afford to buy certain fertilizers any longer. For example, 100% of the respondents said that

they eliminated potassium fertilizers from their fertilization program. Moreover, the system of distributing subsidized fertilizers to smallholders by the agricultural cooperatives seems sometimes problematic, as a smallholder stated.

"The Ministry of Agriculture often delays distributing the subsidized fertilizer. Sometimes I receive my quota three or four weeks later than the recommended growing period". *Smallholder, Interview No. 13*

The interviewed smallholders said that the increase in the wheat production costs massively deteriorated their profitability and that there is little incentive for them to sell their wheat "to the government". This was expressed in a statement by another smallholder:

"Wheat production cost is high, and the government price for wheat is relatively low and not fair to us. We cultivate wheat mainly for our home consumption, and we use wheat straw for animal feed. We prefer to feed our poultry on the rest of our wheat rather than supply it to the government". *Smallholder, Interview No. 7*

Many of the smallholders expressed that they feel unsupported and forsaken by the governmental authorities. In the transect walks, we observed a significant lack of agriculture extension services, and in some villages, extension service units did not exist. Based on the Central Administration for Agricultural Extension in Egypt [22], the total number of agricultural extension staff in Egypt decreased between by 2007 and 2018 from 9658 to 2503. This corresponds to a reduction in the extension work force by 74% within only 10 years. A drastic reduction in governmental support in the form of capacity building and access to production inputs echoes this. Instead, smallholders have to purchase their fertilizers and pesticides from private merchants at relatively high prices. One agricultural extension agent shared his perspective as follows:

"20 years ago, in this extension service unit we were 24 employees helping 1200 farmers, now we are only 2 employees. How can we serve all these farmers? The government stopped employment in this unit in 1995." *Agricultural extension agent, Interview No. 3*

Furthermore, smallholder farming is a physically arduous profession that often only generates low and fluctuating income. The increasing availability of more attractive and profitable job alternatives, especially for younger and educated people, led to a significant shortage of agricultural labor in many parts of Egypt. This is mirrored in the following statement:

"I have two sons. One is working in Alexandria city and the other one is working as a tok-tok driver. The last one rather prefers to pay me the wage of a field worker than helping me in the field. He always says the work in the field is more exhausting, and he makes better money as a tok-tok driver". *Smallholder, Interview No.10*

### 3.4. Percieved Changes in Attributes Affecting Wheat Production between 2000 and 2020

This part of the empirical qualitative data collection was aimed at understanding wheat growers' perceptions of changes in attributes affecting wheat production between 2000 and 2020. We applied a five-point Likert scale and asked smallholders to indicate their perception of change (much worse = 1, worse = 2, same = 3, better = 4, and much better = 5).

The data show that wheat-growing smallholders perceive a significant deterioration of the extension services, the prices for inputs, the access to subsidized inputs, the profitability and the final price between 2000 and 2020. However, interestingly, they also stated that the access to equipment and the access to seeds improved in this period. Market access and fertilizer access were also seen as being either "the same", "better" or "much better" compared to 2000 for the majority of the smallholders (Figure 6).

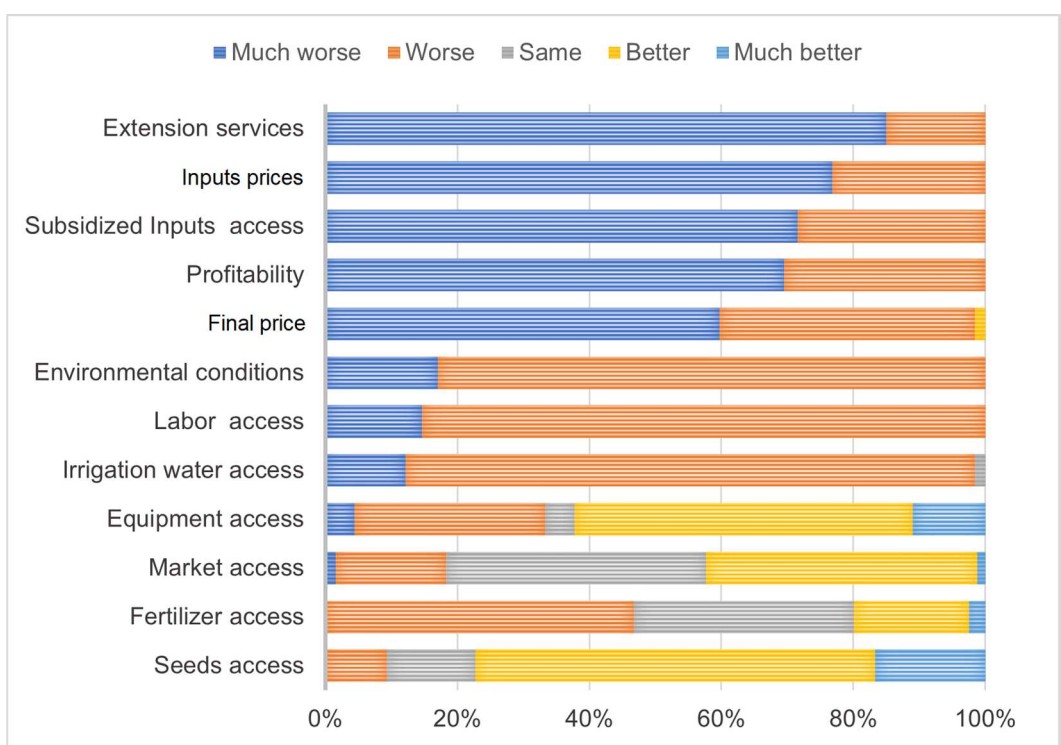

**Figure 6.** Smallholders' perception of the changes in factors affecting wheat production between 2000 and 2020 (Source: field survey 2020/21 growing season).

## 4. Discussion

This study shows that the wheat production of smallholders in Egypt depends on a variety of underlying system-immanent and external factors. While smallholders are the backbone of wheat production in the country, many factors crucially limit their ability and willingness to produce (more) wheat. It is essential to understand smallholders' perspectives and to tackle these factors in a "bottom-up approach" to increase food security [29], to reduce Egypt's dependency on wheat imports and unreliable supply chains and to contribute effectively to the "Egypt Sustainable Development Strategy Towards 2030".

Various studies have investigated the factors that influence smallholders' decisions in crop production and the adoption of agricultural practices [30–33]. All too often, farming systems were categorized into two main categories, subsistence farms and commercial farms, each with seemingly distinct factors and drivers that influence their decision making. In smallholder farming, however, this dichotomy does not reflect reality.

This study investigates the factors that affect wheat-growing smallholders' decisions in semi-subsistence farming systems. The study found that securing their own wheat supply is the main driver for smallholders to grow wheat. However, the study also shows "regulating screws" that have to be turned to make wheat production more attractive for smallholders. The study found that several economic factors, including wheat price, an earlier declaration of the price that smallholders obtain for their wheat when selling it to the public storage silos, and a more efficient and reliable purchase process at the silos can have a strong impact on the smallholders' decisions to grow (more) wheat. Smallholders have a strong cultural attachment to and history in growing wheat. Our data show that wheat-growing smallholders usually have decades of experience in growing wheat. Their parcels are small but blessed with fertile soils. Smallholders know the crop wheat—and the socio-economic system around it—extremely well. From one year to another, they could significantly increase the share of their land on which they grow wheat and could increase their productivity—only if the necessary incentives and inputs are provided, however. With millions of wheat-growing smallholders facing a similar context [34–36], this represents

a huge potential that can and should be used to increase domestic production, improve food security, and thus reduce Egypt's dependency from expensive and unreliable wheat imports.

The results also show that there is a strong integration of wheat production and animal husbandry. The findings are consistent with other studies reporting that smallholders in semi-subsistence farming systems often keep animals for milk and meat production, and as a capital and "assurance" assets, use the manure from these animals as fertilizers, and the residues from the crops to feed animals and for their bedding. This integration contributes to a (more) stable access of smallholders to both food and income [36].

Our results show a strong deterioration of the incentives for smallholders in Egypt to produce wheat in the period between 2000 and 2020. This discouraged smallholders from maintaining (or even increasing) the share of wheat on their land and rather to use their land for growing other competing crops. These findings comply with other studies, for example with [37] who observed a reduction in the land used to cultivate cowpeas in Haiti. In general, our findings are consistent with prior research, which has documented that smallholders in many countries of the Global South faced significant challenges in the last two decades with reduced incentives and chances to grow their crops [14]. In contrast, however, other studies highlighted that smallholders were able to significantly increase their crop production, under given circumstances. For instance, contract farming in which smallholders and buyers both agree in advance on the terms and conditions for the production and marketing of certain products, can provide smallholders with better access to markets and technical assistance, and most importantly, less insecurity and exposure to price volatility though predetermined guaranteed prices. This can provide them with the incentives and resources they need to invest in higher productivity of their farms as reported in [38,39].

The adoption of sustainable agriculture practices is impacted by factors such as access to knowledge and training and access to production inputs confirming to [34–36]. This study reveals that wheat-growing smallholders in Egypt perceive a considerable deterioration of the incentives and their capacities to produce wheat sustainably. Similar findings have been reported in other studies conducted in Egypt, also documenting also that prices for production inputs have risen sharply in recent years [14]. Other studies conducted with smallholders in countries of the Global South, such as in Ethiopia, have also reported concerns that negatively affected smallholders' abilities to access production inputs and to increase their yields [31]. Our study also shows that the distribution of (subsidized) inputs to smallholders is often subject to top-down structures and mismanagement. The findings are consistent with previous studies conducted in Egypt [40].

The findings of our study indicate that wheat-growing smallholders in Egypt majority do not feel supported by the governmental extension services. They perceive the extension services to be inadequate, with limited access to information, training, and technical assistance. In the study area, we observed a significant lack of agriculture extension services. This mirrors a general trend in which the total number of agricultural extension staff in Egypt decreased between by 2007 and 2018 from 9658 to 2503 [30]. The massive reduction in extension staff and services has hampered the adoption of new technologies and adequate sustainable practices, resulting in reduced yields and lower incomes for smallholders. These findings are consistent with previous studies [30], which showed that extension services are often inaccessible to smallholders, who, however, often produce the bulk of the food and constitute the majority of the rural population.

The significant impact of agricultural policies on smallholder crop production and productivity is shown in many studies [34–36,41]. We can recapitulate that policies implemented by the Egyptian governments since 2000 rather aimed to support the production of export cash crops like citrus and the expansion of agricultural land to the "new lands". At the same time, they rather neglected the domestic wheat production by smallholders and weakened the productivity and agricultural development in the "old lands". In conjunction with a population increase from about 70 million in 2000 to about 105 million in 2023 [12]

this led to more and more dependency on wheat imports, of which 85% came from Russia and Ukraine in 2021, before the Russia–Ukraine war. The war triggered skyrocketing prices for wheat on the international markets that, coupled with the depreciation of the Egyptian currency, provoked economic, social and food crises. Over the last few decades, highly fertile "old lands" along the River Nile were increasingly used unproductively and were shifted to other agricultural and non-agricultural purposes. Millions of hectares have been converted into land for housing and infrastructure. Other areas formerly used for growing wheat are now used to produce clover. This all brought along losses in the agriculture production revenue, especially with the traditional crops such as wheat [42,43]. Another challenge shown in the findings of our study is the relatively high age of the wheat-growing smallholders. The mean age of the smallholders in the case study is 56 years. The mean age of the whole population in Egypt is, however, only 24.6 years [12]. This over-aging of the people who produce the bulk of the wheat in Egypt is highly problematic. It threatens not only the goal of achieving more domestic wheat self-sufficiency, but the future of the agricultural sector in Egypt as such. Thus, there is a significant need for agriculture policies that make the agricultural sector more lucrative and attractive to the younger generation.

Given the long-term neglect of the wheat-producing smallholders and the short-term impacts of the Russia–Ukraine war, food security in Egypt is at high risk. In the medium term, the millions of wheat-producing smallholders in Egypt can play a critical role to prevent millions of Egyptians from (more) poverty and hunger and to avert socio-economic instability, unrest and political conflicts in Egypt. Accordingly, to be more effective and sustainable, agricultural policies need to be based upon the particular needs of wheat smallholders, mainly in the "old lands". Egypt has very limited cropland, so the policies should aim to increase wheat yields per unit of land in a "vertical approach" mainly in the "old lands"—which can, however, go in parallel with a "horizontal approach" of reclaiming "new lands" for wheat production.

Given the relatively high fertility of soils in the "old land", the mostly sufficient availability of water, and—not to forget—the long experience and strong cultural attachment of millions of smallholders in and to wheat production, more wheat could be produced in a relatively short time by only adjusting the agronomic practices such as irrigation and inputs managements [44]. This, however, needs substantial and sustained investments and support that should include: (1) providing wheat-growing smallholders with technical support and knowledge through a (re-)strengthened agriculture extension system, (2) paying smallholders higher prices for their wheat at public silos, (3) providing wheat-growing smallholders with subsidized inputs on time, (4) declaring the wheat prices before the growing season, and (5) building a more effective domestic wheat collection system. Each of these activities will be costly. However, when these investments are being made, billions of USD can be saved every year through lower wheat imports, and more socio-economic benefits will be generated through higher domestic economic growth.

## 5. Conclusions

This study highlights the importance of understanding smallholders' perspectives and underlying factors that influence their ability and willingness to produce (more) wheat. Drawing from nine months of empirical, mixed-method field research in the Nile River Delta, the study found that securing their own food needs and gaining animal fodder are the main factors that influence smallholders' wheat production. Economic factors, including the wheat price declared by the Ministry of Supply and Internal Trade, the time of the price declaration, and the costs of production inputs such as fertilizers, pesticides, and seeds, can, however, have a significant impact on the land share on which smallholders grow wheat, as well as on their productivity, and ultimately on the total wheat production in Egypt. The study reveals the negative impacts of long-term policies and a rundown of the governmental extension system over decades on smallholder decision making, leading to decreased land under wheat and lower productivity. Most smallholders cannot afford production inputs and perceive a lack of access to knowledge and information. The

access to knowledge and training has to be guaranteed through an effective extension service. Smallholders need sufficient access to subsidized agricultural inputs to meet the essential nutritional requirements for their crops and achieve a significant improvement in productivity. The study also showed that the factors influencing farmers' decisions to grow wheat or implement innovative practices vary across different areas within the same region. Therefore, it is essential to integrate smallholders' perspectives and needs into the making of targeted policies and interventions, to adapt the tax and subsidy systems with regard to wheat production, to (re-)strengthen the agricultural extension, and to implement them bottom-up accordingly to increase food security and domestic self-sufficiency and to effectively contribute to the "Egypt Sustainable Development Strategy Towards 2030".

**Author Contributions:** Conceptualization, A.A., M.B. and T.S.; analysis, A.A.; data curation, A.A.; writing—original draft preparation, A.A.; writing—review and editing, A.A., M.B. and T.S.; supervision, M.B. and T.S. All authors have read and agreed to the published version of the manuscript.

**Funding:** This study was funded through the Right Livelihood College (RLC) Campus Bonn by the German Academic Exchange Service (DAAD), grant number 91757798, and the Hermann Eiselen Programme of the Foundation Fiat Panis, grant number 91757798. The APC was funded by the University of Bonn.

**Institutional Review Board Statement:** This study was approved by the Institutional Review Board at the Research Ethics Committee, Center for Development Research (ZEF), the University of Bonn, (protocol code: Ethical Clearance Form for BIGS-DR doctoral program—Guidance Note. on 15 September 2020. All participants provided informed consent prior to participating in the study. The study protocol was designed for studies involving humans to minimize any potential risks to participants, and their confidentiality was protected throughout the research process.

**Data Availability Statement:** The data presented in this study are available on request from the corresponding author.

**Acknowledgments:** We gratefully acknowledge the funding from the Right Livelihood College (RLC) Campus Bonn by the German Academic Exchange Service (DAAD), and the Hermann Eiselen Programme of the Fiat Panis Foundation. We are thankful to our Egyptian partner SEKEM, who received the Right Livelihood Award in 2003 for their support during the field research.

**Conflicts of Interest:** The authors declare no conflict of interest.

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
