# Peer review of "Wheat Farmers’ Perception of Constraints and Their Adaptive Capacity to Changing Demands in Egypt"

_agriculture, doi:10.3390/agriculture13081554_

Round 1

Reviewer 1 Report

The study deals with a very important topic concerning the conditions for ensuring food security in Egypt. As a result of the war between Russia and Ukraine, the traditional channels of food supply have been broken down, and the supplies of wheat used to produce bread subsidized by the Egyptian government have been interrupted. In connection with the above, the study undertook to analyze the production possibilities of small farmers producing in the Nile basin in Egypt. The subject matter taken up in the study is international due to Egypt's dependence on wheat imports, the study should be published, it will be of interest to a wide audience - politicians, entrepreneurs and consumers.

The following arguments lead to publishing the study, after the authors have taken into account some of the following comments, especially the comments in points 5 - 7.

1. The research questions were developed correctly.

2. The methodology of the conducted research was properly selected to achieve the selected research goals.

3. The research method was clearly described in the study.

4. Appropriate statistical methods were used to describe and interpret the studied phenomena.

5. The selection of sources is quite modest, however, the discussion is of a scientific nature, there are no references to studies conducted using comparable research methodology.

6. The text was written in a professional and understandable language for the recipient. However, the text requires careful reading for typos and grammatical errors, e.g. line 484: This is followed by policy factorsthe area?

7. Conclusions should refer to the research carried out. Line 700 raises objections: "Smallholders need subsidized production inputs if they should significantly increase their productivity." It is doubtful whether this conclusion was derived on the basis of the conducted research or is it a nugget conclusion of the authors not supported by research or literature studies?

Author Response

Thank you for forwarding this helpful review report. We are most grateful for
the time you spent providing suggestions on how to improve our paper. In our revision, we have tried to address your suggestions as well as possible, as specified in detail in the attached report.

with kind regards,

Reviewer 2 Report

In the introduction, the study deals a lot with the importance of wheat as a food, the domestic situation of wheat production, market conditions, export vulnerability, but relatively little with the situation of small producers in Egypt. This should be supplemented by a more detailed presentation of the general production conditions of small-scale wheat-growing farms in Egypt (e.g. applied technology, yield averages, procurement of inputs and typical supply chains, storage methods and logistics, availability of support forms, available credit schemes, etc.). At the end of the introduction chapter, it would have been good to formulate some hypotheses related to the research work.

More could have ben done with the likert-scale type questions on the quantitative survey part. Not only the simple distribution should have been presented, but also e.g. calculate average values and standard deviations, perform correlation tests, etc.

The thorough primary data collection by the authors is highly commendable. The study deals with a real problem and provides valuable information for Egyptian agricultural decision-makers in addition to representatives of the scientific sphere.

Author Response

(The authors gave the same response as above.)

Reviewer 3 Report

See attached file for comments.

Well-written, easy to read, but a few typos must be corrected.

Author Response

(The authors gave the same response as above.)
